# Clinical Outcomes after Surgical Resection Combined with Brachytherapy for Uveal Melanomas

**DOI:** 10.3390/jcm11061616

**Published:** 2022-03-15

**Authors:** Isabel Relimpio-López, Antonio Manuel Garrido-Hermosilla, Francisco Espejo, María Gessa-Sorroche, Lourdes Coca, Belen Domínguez, María Jesús Díaz-Granda, Beatriz Ponte, María José Cano, Enrique Rodríguez de la Rúa, Francisco Carrasco-Peña, Carlos Míguez, Jonathan Saavedra, Antonio Ontanilla, Carlos Caparrós-Escudero, Juan José Ríos, José Antonio Terrón

**Affiliations:** 1Department of Ophthalmology, Virgen Macarena University Hospital, 41009 Seville, Spain; gaherfamily@hotmail.com (A.M.G.-H.); dr.franciscoespejo@gmail.com (F.E.); mariagessasorroche@gmail.com (M.G.-S.); lucoca79@yahoo.es (L.C.); belend-g@hotmail.com (B.D.); doctoradiazgranda@hotmail.com (M.J.D.-G.); bepontezu@hotmail.com (B.P.); mariajez@telefonica.net (M.J.C.); enrique.rodriguezruz.sspa@juntadeandalucia.es (E.R.d.l.R.); 2Department of Radiation Oncology, Virgen Macarena University Hospital, 41009 Seville, Spain; francisco_cp84@hotmail.com (F.C.-P.); carlos.miguez.sspa@juntadeandalucia.es (C.M.); jonathan_saavedra_@hotmail.com (J.S.); 3Department of Anesthesiology, Virgen Macarena University Hospital, 41009 Seville, Spain; antonio.ontanilla.sspa@juntadeandalucia.es; 4Department of Radiology, Virgen Macarena University Hospital, 41009 Seville, Spain; ccescudero@gmail.com; 5Department of Pathology, Virgen Macarena University Hospital, 41009 Seville, Spain; jjrios@us.es; 6Department of Radio Physics, Virgen Macarena University Hospital, 41009 Seville, Spain; jose.terron.sspa@juntadeandalucia.es

**Keywords:** uveal melanoma, brachytherapy, exoresection, surgery

## Abstract

Currently, brachytherapy is the most commonly used therapeutic approach for uveal melanomas. Surgical resection by means of endoresection or exoresection is an alternative approach. The present report recounts our experience over 15 years in the treatment of uveal melanoma using a combined approach of resection surgery with brachytherapy. This is a single-center observational retrospective cohort study in which we describe clinical outcomes, complications and survival in 35 cases of melanoma of the iris or the ciliary body after a combination of surgery and brachytherapy or brachytherapy alone. Local treatment of the tumor was successful in all cases with surgery and brachytherapy. The most frequent complications were scleromalacia, bullous keratopathy, retinal toxicity, cataracts, hypotonia, and photophobia. There were three cases of recurrence, all of which were found in the group of patients who had received brachytherapy alone, and in one case we had to perform a secondary enucleation due to tumor growth after brachytherapy. At present, only one patient has died during follow-up due to liver metastases six years after the start of treatment. In carefully selected patients, this approach can be effective and safe, as long as a close follow-up is carried out after surgery.

## 1. Introduction

Uveal melanoma is the most common primary intraocular malignancy and remains fatal. Currently, brachytherapy, using isotopes of iodine-125 (I-125) and ruthenium-106 (Ru-106), is the most commonly used therapeutic approach for uveal melanoma, with a local control rate in the range of 88–98% at five years [1], and it is equivalent to enucleation in terms of prevention of systemic spread and death [2]. Consequently, plaque brachytherapy remains the method of choice to control primary intraocular tumors [3,4].

Surgical resection of uveal melanomas is an alternative approach. Currently, there are two surgical resection techniques: transscleral resection or “exoresection” using a partial lamellar sclerouvectomy, and “endoresection” via a pars plana vitrectomy. While exoresection is more suited to anteriorly placed tumors with ciliary body or iris involvement, endoresection is better for posteriorly located tumors without ciliary body involvement. Interestingly, both approaches are suitable for large tumors >8 mm thick [5]. In this context, a surgical approach holds certain advantages. By removing the tumor burden from the eye, we obtain histopathological and cytogenetic information about the tumor and the complications associated with the so-called toxic tumor syndrome are avoided. However, both types of surgical resection are challenging surgical procedures, carrying the risk of early and late postoperative complications [6,7].

In our hospital, we have conducted a multidisciplinary approach to iris and ciliary body melanoma by combining resection surgery with brachytherapy in a few of the initial cases described [8]. The objective of this paper is now to describe our experience over 15 years in the treatment of iris and ciliary body melanoma using a combined approach of resection surgery with brachytherapy or brachytherapy alone. Based on our experience, we aim to describe the best surgical approach, complications in the follow-up and clinical outcomes. The results show how challenging this approach is and we hope it will help to establish the best approach for the treatment of these tumors.

## 2. Methods

This is a single-center observational retrospective cohort study describing our experience since 2010 in the management of anterior uveal melanomas using a combined approach of resection surgery and brachytherapy. The candidate patients were those diagnosed with a melanoma of the ciliary body or iris whose size, plus a safety margin of 1–2 mm, does not exceed one quadrant of the eye, regardless of the height of the lesion. Patients with scleral invasion or systemic metastases were excluded from the procedure.

We evaluated the size and the resection of the lesion with ultrasound biomicroscopy (Figure 1) and magnetic resonance. T staging classification was carried out according to the American Joint Committee on Cancer classification [9].

### 2.1. Procedures

The surgical approach varied according to the characteristics of each case. However, the surgery and the follow-up of all the cases were performed by the same surgeon to reduce the variability in the technique and follow-up. In small tumors confined to the ciliary body or the iris invading the iridocorneal angle, we performed an iridocyclectomy (iris+ ciliary body) by means of exoresection. To avoid the sclera drying out and thereby achieving a lower degree of long-term scleromalacia, we initially applied a novel technique with an external and an internal cap that could be coapted, thus avoiding the filtration of aqueous humor in the postoperative period which may lead to sclerectomy (Figure 2). However, the incision area may be observed due to the brachytherapy applied at the end of the surgery [10].

In larger tumors and in those that fundamentally affected the ciliary body with posterior extension, we performed a sclerouvectomy (resection of sclera+ uvea). In these cases (Figure 3), the lamellar scleral flaps were limbus based or fornix based depending on whether the superior and inferior lateral rectus muscles were involved (Figure 4). Therefore, we considered performing endoresection with the endoscope (Figure 5a) two years after the brachytherapy in patients who required sclerouvectomy, especially in those of greater size and greater complications.

The tumor was marked before and after the flap (Figure 4a,b) by transillumination, and with a knife we cut the sclera to proceed with the endocautery and transscleral diodopexy to prepare the uvea for exoresection (Figure 2b). The uvea was cut with microscissors, Wescott scissors and vitreotome (Figure 3c), using hypotensive anesthesia in every case to avoid bleeding. Cataract surgery was performed after the scleral flaps, and we attached support rings to the sector to avoid photophobia, and Cionni (Figure 4b) rings to anchor them to the sclera in case of a possible extensive zonule disinsertion.

We attempted to detach the tumor from the hyaloid, keeping it intact to work in the closed chamber (Figure 3a) closing the flap before completing the exoresection in order to avoid hypotonia. (Figure 3b). To prevent this affecting the iris, we resected the anterior part of the tumor with vitreous microscissors (Figure 3c), trying to keep a scleral spur to reduce astigmatism (Figure 2d), making sure, where possible, not to leave melanoma cells.

Complete vitrectomy (23 g) was performed in almost all the patients, but without peeling back the internal limiting membrane, which may hinder the formation of epiretinal membranes in some cases. Silicone oil (1300 cs) was used in patients with a posterior vitrectomy with risk of hypotonia.

In cases of lifting during exoresection, the peripheral retina was treated with extrascleral diodopexy (Figure 5b,c), endolaser or cryotherapy. With both procedures, in some patients we added equine pericardium, autologous tenon and autologous and heterologous sclera transplantation in punctual perforations during surgery and severe scleromalacias (Figure 4f) and in scleromalacias (Figure 5d). We also treated the tenon with conjunctival zetaplasty (Figure 5e). The brachytherapy plaque was placed on top of the sclera or cornea, depending on the surgical bed, after surgery in both types of resections, covering it with the conjunctiva if possible (Figure 2f). One interesting finding we observed was that pigmentary lesions appeared under the conjunctiva, simulating slecromalacia or tumor invasion, which in fact resulted from macrophages migrating from the pigmentary epithelium. This was observed in patients with brachytherapy, more frequently with Ru-106 (Figure 5f) [11].

For brachytherapy, the prescription dose was 100 Gy at 2 mm depth for patients who underwent sclerouvectomy and brachytherapy in the same operation [12]. When brachytherapy was performed before the sclerouvectomy, the prescription was 85 Gy to the tumor apex [13]. When the prescribed dose level was reached, the plaque was removed in the operating room. Treatment time, indicated in the dosimetry set by the medical physicist and approved by the radiation oncologist, was never exceeded or reduced. If the plaque was placed over the cornea, after plaque removal, an epithelial ulcer could appear. This ulcer was dealt with without complications with one week’s topical treatment.

### 2.2. Follow-Up

We conducted a close follow-up of these patients. During the six months after surgery, they were reviewed on demand, on average every week. From then on, the follow-up was programmed on average every two months, although this varied widely depending on the complications and clinical results. Follow-up was conducted indefinitely, until death or lost to follow-up. Vital status was checked on 31 December 2021. During the follow-up, according to the recommendations by Schulze-Bonsel K, et al. [14], the following factors were used to estimate visual acuity: finger count: 0.0130, hand movement: 0.0050, view of bulks: 0.0025, light perception: 0.0020, no light perception: 0.0010, enucleation: 0.0000. This surgery generates a high level of astigmatism and the visual acuity measurements were taken without the refractive errors being corrected, since we focused on the viability of the eye rather than the visual acuity.

### 2.3. Ethics

This study followed the recommendations of the Declaration of Helsinki for studies with human beings issued by the World Medical Association. In compliance with the provisions of Organic Law 15/1999, dated 13 December, on the Protection of Personal Data, subsequently updated according to Organic Law 3/2018, dated 5 December, on the Protection of Personal Data and Guarantee of rights (LOPDGDD) and its associated regulations, the patients’ personal data were kept strictly confidential and only the clinicians involved in treating the patient were given access to them. No personal data were stored on the database that allowed the patient to be identified. The treatments performed are included in the medical unit’s range of available treatments, which are selected to best meet each patient’s needs. The patients signed an informed consent form prior to each of the procedures described where they were informed of the purpose of the operation and its possible complications.

### 2.4. Statistical Analysis

The statistical analysis was performed using the IBM SPSS Statistics program (IBM Corporation, Armonk, NY, USA), version 26.0. As this was a descriptive analysis, we included the univariate descriptive statistics of the patients’ baseline characteristics and the characteristics of the clinical results and intraoperative and follow-up complications. The qualitative variables are shown with the absolute and relative frequencies observed for the categories referring to the total number of patients (unless otherwise specified). Continuous variables are described as mean and standard deviation. Cases were compared in terms of the first therapeutic approach comparing those initiating therapy with Brachytherapy + surgery versus those with initial brachytherapy alone. These comparisons were made using the chi-square or Mann-Whitney tests, depending on the nature of the variable. Statistical significance was set at 0.05.

## 3. Results

During the study period, 35 cases were included, of which 26 received a combined surgery with intraoperative brachytherapy and nine received brachytherapy alone as the first approach. The description of these cases is summarized in Table 1. This was a cohort with an average age in their fifties (from 23 to 83 years of age), with melanomas affecting mainly the iris and ciliary body, most of which were small, between 1 and 3 mm in height and less than 5 mm in basal diameter. The cases included three (8.6%) epithelioid, 23 (65.7%) fusiform A, 4 (11.4%) fusiform B, and 5 (14.2%) cases with no histology. Information on the brachytherapy applied is summarized in Table 2. The doses in the optic nerve and macula were almost zero, and in most cases were treated with Ru-106, since the beta radiation of this isotope has a short range. With I-125, it also has a low value since the distance between the tumor and the macula or the optic nerve is great. 

Patients had a follow-up after six years on average (Table 3). During the follow-up, visual acuity decreased in the first six months, but remained stable thereafter with some differences between both treatment groups. The main complications that arose after the start of treatment are summarized in Table 3. The most frequent complications, in over 10% of the patients, were scleromalacia, bullous keratopathy (Figure 5d,e), retinal toxicity, cataracts, hypotonia, and photophobia. Of the 14 cases with scleromalacia, 8 (57.1%) were mild and did not require operation. Cataracts were all solved by phacoemulsification during the first or second surgery when the exoresection was performed. Retinal toxicities were managed with dexamethasone. We observed epiretinal membranes in three patients, with vitrectomy and peeling of the internal limiting membrane to avoid recurrence. Half of the hypotonias were resolved without treatment. The photophobia was mild and did not require treatment. Two patients underwent endothelial keratoplasty with anatomical but not functional success and required penetrating keratoplasty. One patient presented endophthalmitis due to corneal abscess in the graft and was enucleated at their local hospital as they were unable to travel to our hospital because of the distance. There were 3 cases of relapse and in one case we had to perform a secondary enucleation. At present, only one patient has died during follow-up due to liver metastases, six years after the start of treatment.

## 4. Discussion

This paper describes the experience of a reference hospital in uveal melanoma surgery, and the results of an alternative approach to conventional brachytherapy as the initial approach. Our results show that a strategy based on surgery and combined with brachytherapy achieves optimal results, with different degrees of complications and a good long-term survival rate.

Ocular melanoma accounts for 5% of all melanomas, making it the most common primary malignant intraocular tumor in adults. 85% of ocular melanomas originate in the uvea, with the most common location being the choroid (80% of the total), followed by the ciliary body (12%) and iris (8%). Brachytherapy is the most commonly used form of radiotherapy for uveal melanoma, and with the use of I-125 and Ru-106 isotopes, it achieves a local control rate in the range of 88–98% at five years [1,15]. Despite its results, radiation retinopathy is the most common ocular side effect after brachytherapy for uveal melanoma [10]. Its incidence is reported to occur in 10–63% of eyes, with higher rates associated with larger tumors, higher radiation dose, and posterior location [16,17]. In this context, the correct placement of brachytherapy plaques improves the precision of the treatment of uveal melanomas. A number of different techniques are available to achieve this. Transpupillary transillumination is mandatory to place the brachytherapy plaque in all uveal melanomas, and it is easy to mark the limits in iris and ciliary body melanomas due to the location. High-frequency ultrasonography or ultrasound biomicroscopy is used to measure the height and base of ciliary body melanomas before surgery.

Partial lamellar sclerouvectomy was invented over 30 years ago [18,19] in order to remove melanomas that involved the ciliary body or the choroid, while leaving the outer portion of the sclera and the overlying sensory retina intact. Additionally, there are a few circumstances in which surgery should be the first option [5]. Removal is indicated for uveal melanomas of indeterminate pathology that present suspicious features, such as localized pigment seeding, prominent vascularity, and increased thickness and size, where fine-needle aspiration biopsy is possible, but it can give us false negatives due to the lack of material. Larger ciliary body and choroidal melanomas require high radiation doses directed towards the tumor and surrounding tissues, resulting in potential radiation complications, and surgical resection is therefore believed to be safer under these circumstances [20]. In many countries, plaque radiotherapy or proton beam radiotherapy for iridociliary melanomas may not be available. Additionally, growing necrotic iridociliary melanomas can lead to an increase in intraocular pressure [21]. In fact, angular invasion and increased tension provides us with a clue to the malignancy of the lesion. Thus, removal is an option and sometimes necessary to control the intraocular pressure and intraocular inflammation. Interestingly, some iridociliary tumors with rapid documented growth need to be surgically excised to restore the intraocular structure and prevent complications. In fact, in posterior pole tumors, endoresection may conserve the central vision or the temporal field in cases where radiotherapy would be expected to cause optic neuropathy [22]. The radiation dose is also considerably lower and therefore the surface may suffer less.

Isolated brachytherapy is always a valid, less aggressive alternative in the immediate postoperative period, and requires fewer later check-ups. The limitations that we found with brachytherapy treatment alone, which is what drove us to pioneer this technique, were two-fold: first, the possible lack of biopsy due to scarce material, and secondly, the lack of certainty that there will be no local recurrence because the lesion does not disappear completely. However, after analyzing our results and complications and knowing that small iris tumors have a very low metastasis rate, we were forced to reconsider the possibility of offering surgery as a complementary option [23]. Since isolated brachytherapy leads to fewer complications in the immediate postoperative period, we could start with brachytherapy with the possibility of surgery if there was long-term problem. In the case of ciliary body tumors, which have a worse prognosis, this surgery could be via an endoresection (Figure 5a) with endoscopy within a period of two years after the brachytherapy, when it is expected that all the effects of the radiation will have been evident. This would allow us to avoid scleromalacia and hypotony wherever possible, due to the high risk of perforation involved. This endoresection can be completed with the endoscope so as not to leave traces of malignant tissue. In addition, since the entire area would already be irradiated, there is less bleeding, and cell dissemination would not be considered such a high risk.

Another option is to combine brachytherapy and surgery simultaneously. In this way, the surgery slightly lowers the radiation dose of brachytherapy. Additionally, brachytherapy helps control bleeding in the immediate postoperative period. Brachytherapy can therefore be used as an adjuvant treatment to treat the source of possible remaining microscopic disease and reduce the probability of local recurrence. This treatment makes it possible to maintain the eyeball and preserve eyesight with good local control without jeopardizing the possibility of performing other required treatments in the future.

This is a type of surgery in which the management of the anterior and posterior pole must be performed by an expert team and not all patients may benefit from it. Depending on the area in which they occur, uveal tumors will be handled differently, due to the complexity of the surgery and the possible complications that may arise. However, since they are anterior, and of small to medium size, they can be diagnosed earlier, especially if they affect the iris, which is easily seen with the slit lamp. Since ultrasound biomicroscopy is not a standard technique in an ophthalmologist’s routine examination, tumors of the ciliary body are usually diagnosed when larger, which may worsen the prognosis. The radiation doses that reach the optic nerve and macula are lower, so depending on the size, it will be necessary to evaluate with all the data available whether exoresection will be beneficial for the patient [24].

There are a number of considerations here that must be taken into account. Firstly, this is a single-center retrospective analysis. Secondly, the tumors treated in this report were small to medium in size, without local or distant infiltration. In addition to this, close postoperative monitoring represents a key factor in the outcome of treatment. In our cohort, the patients were followed very closely, which enabled us to identify and treat complications early. This implies that proposing this surgery to patients who cannot attend check-ups easily would be a bad option because they need a close follow-up. A close postoperative follow-up period managed by committed, expert personnel is fundamental because there is a wide range of complications in both anterior and posterior poles. It is also important to note that the possibility of performing a biopsy with better samples would allow us to obtain the additional benefit of performing a gene expression profile.

Although it has been described that ciliary body tumors have a poor prognosis, this was not born out in our experience. Interestingly, previous work has also shown similar percentage of relapse or mortality. Damato et al. followed 52 endoresections for choroidal melanoma and no patients developed definite local tumor recurrence and only one died of metastatic disease 41 months postoperatively [25]. Our recurrence or mortality data are low, which could be related to a combination of surgical experience and strict safety measures.

Severe scleromalacia were solved with a heterologous corneoscleral endopatch so as not to cut the conjunctiva to avoid complication with blinking in the immediate postop. We found no differences in complications in terms of the histology of the tumors. The only death occurred in a fusocellular type, which has a better prognosis. In any case, deaths are usually associated with distant disease, mainly due to liver metastases. Notably, the treatment presented here is local, and, in theory, does not affect distant disease. Prognostic information should therefore be based on clinicopathologic variables, including baseline tumor length (<12 mm would present better prognosis), ciliary body involvement, thickness, number of mitoses and vascular alterations [26].

Endoscopic endoresection was a useful method of exploring the ciliary body accurately. This is a new application and was successfully used here. The consistency of the tumor after brachytherapy is higher and bleeding is low, which helps us to perform the endoresection with good visualization. We used 20 g because of its greater consistency and because it was previously irradiated. In our view, it can provide a useful alternative in patients where exoresection can be ruled out because of the large number of medical appointments required in the immediate postoperative period. In exoresections, performing a complete posterior pars plana vitrectomy or not it depends on the size of the ciliary body tumor.

Cataracts were all solved by phacoemulsification in the first operation when the exoresection was performed, except in two cases in which the intervention was postponed to a second operation. The intervention was programmed in this way because the latest reports of the International Commission on Radiological Protection [27], specify a limit of 0.5 Gy, compared with the previous threshold doses for visual-impairing cataracts of 5 Gy for acute exposures and >8 Gy for highly fractionated cataracts [27]. The lens doses in our treatments were over the limits. Cataract intervention at the same time, when possible, prevents the patient from having to undergo a new operation in the same zone. This complication occurred in all cases with brachytherapy alone (except the last four cases which had a shorter follow up) and in two cases in which cataract intervention could not be performed in the first operation.

Loss of media transparency due to bullous keratopathy is present in several patients three to five years after surgery, with no significant loss of endothelial cells during surgery and no significant edema in the immediate postoperative period. We consider that this loss of transparency may be due to limbal insufficiency, but it is surprising that they do not have recurrent ulcers and respond well to corneal transplantation with a speed of re-epithelialization within the normal range.

## 5. Conclusions

In conclusion, the present report highlights the benefits of a combined approach for uveal melanoma with a combination of brachytherapy and surgery. With the experience gained, considering the complications and the strict follow-up visits, we would be inclined to start with endoresection techniques after treatment with brachytherapy for ciliary body tumors, whereas exoresection followed by brachytherapy or brachytherapy alone would suit small tumors better. In carefully selected patients, this approach has shown to be effective and safe, as long as a close follow-up is conducted after surgery. Accordingly, it should be offered to candidate patients as another option, and there are a number of clinical scenarios in which it may be the most suitable approach. Altogether, the availability of different approaches will be of added value in order to select the best approach for specific patients with this neoplastic eye disease.

## Figures and Tables

**Figure 1 jcm-11-01616-f001:**
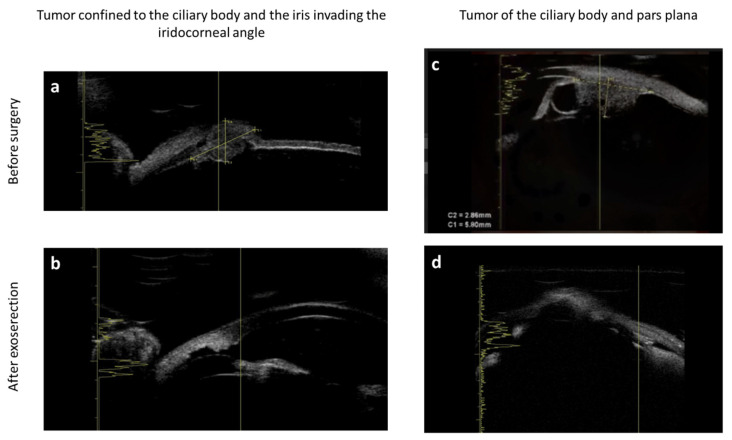
Ultrasound biomicroscopy of iris and ciliary body tumors. (**a**,**b**) Iris tumor invading angle; (**c**,**d**) ciliary body and pars plana tumors.

**Figure 2 jcm-11-01616-f002:**
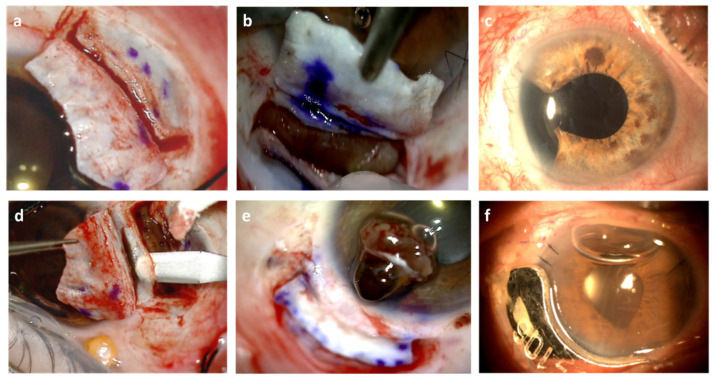
Images of the surgical field showing: (**a**) double flap in smaller tumors to avoid thinning of the sclera; (**b**) double flap, with diodopexia applied; (**c**) pupiloplasty+ sector ring to avoid photophobia; (**d**) if technically possible, and no significant invasion is likely, we try to maintain the scleral spur to reduce astigmatism; (**e**) exoresection of iris melanoma with spur invasion; (**f**) Ru-106 over the surgical exoresection bed.

**Figure 3 jcm-11-01616-f003:**
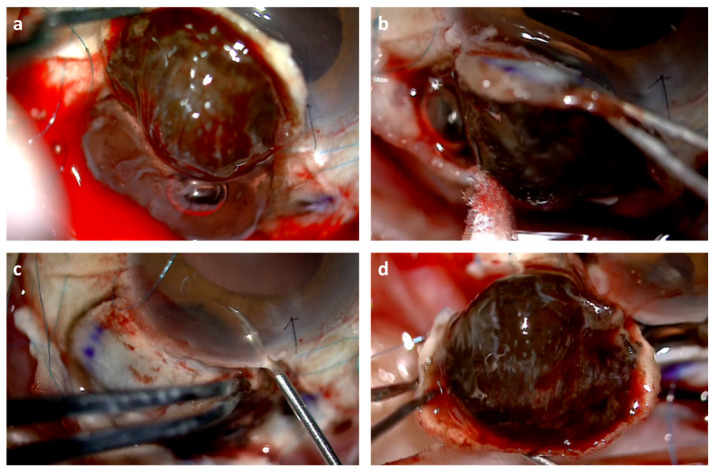
Exoresection procedure showing: (**a**) exoresection keeping the hyaloid intact; (**b**) closing the flap before tumor resection: the bubble can be seen that indicates that the hyaloid is still intact; (**c**) anterior resection with microscissors; (**d**) resection of complete specimen.

**Figure 4 jcm-11-01616-f004:**
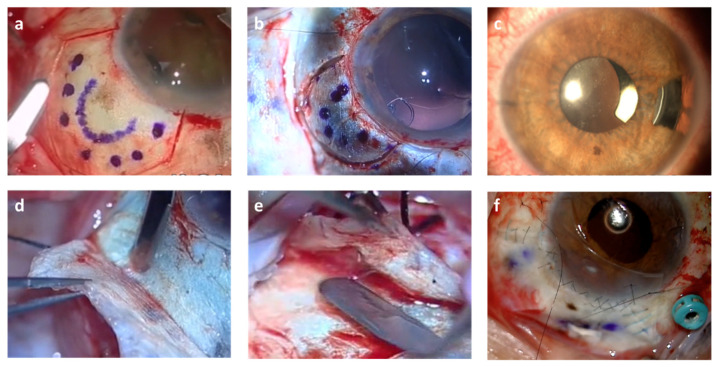
Details of the different surgical techniques, showing: (**a**) ciliary body tumor not involving any muscle with limbus base flap; (**b**) Cionni ring placed before exoresection. (**c**) image of a local exoresection with a Morcher ring with a sector to avoid photophobia; (**d**) vertical fibers in sclera belonging to the muscle tendon and dissection level deepened in sclera to avoid affecting the muscle belly and perforating the flap; (**e**) muscle belly observed after perforating the sclera, fornix based flap; (**f**) autologous sclera and dura mater micropatch.

**Figure 5 jcm-11-01616-f005:**
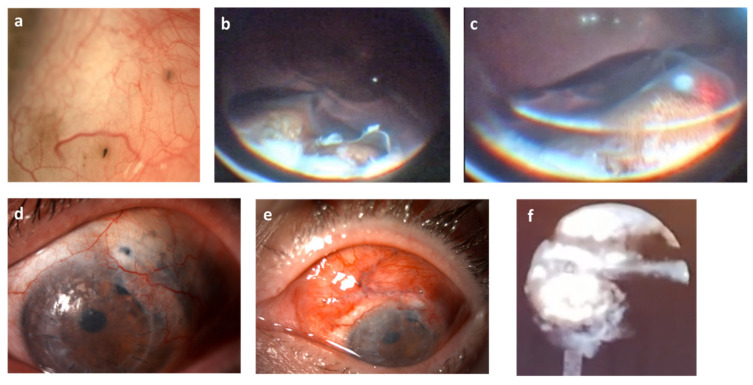
Examples of different techniques after surgery, showing (**a**) migration of macrophages from the pigment epithelium 2 years after treatment with ruthenium-106 plaque.; (**b**,**c**) extrascleral diodeopexy in peripheral rupture after exoresection in tumor previously treated with brachytherapy; (**d**) punctate scleromalacia; (**e**) treatment with tenon patch and conjunctival zetaplasty; (**f**) endoresection image with ocular endoscope.

**Table 1 jcm-11-01616-t001:** Description of the 35 cases included according to the initial approach.

Variable	Total Sample * (*n* = 35)	Brachytherapy + Surgery * (*n* = 26)	Brachytherapy Alone * (*n* = 9)	*p* Value ^¶^
Gender (male)	16 (45.7%)	12 (46%)	4 (44.4%)	0.62
Age (years)	58.0 (15)	57.3 (15.5)	60 (13.9)	0.86
Operated eye (right)	13 (37%)	9 (34.6%)	4 (44.4%)	0.69
Anatomical extension:				
Iris	26 (74.3%)	24 (92%)	2 (22%)	<0.001
Ciliary body	28 (80%)	20 (77%)	8 (89%)	0.64
Choroid	2 (5.7%)	1 (4%)	1 (11%)	0.45
Anatomical extension groups:				<0.001
Iris only	6 (17%)	6 (23%)	0 (0%)	
Iris and ciliary body	20 (57%)	18 (69%)	2 (22%)	
Ciliary body only	7 (20%)	1 (4%)	6 (66.7%)	
Ciliary body and choroid	1 (3%)	1 (4%)	0 (0%)	
Choroid only	1 (3%)	0 (0%)	1 (11%)	
Radial size (mm)	6.6 (2.3)	6.1 (1.6)	8.2 (3.3)	0.17
Circumferential size (mm)	7.1 (2.4)	6.5 (1.8)	9.0 (3.2)	0.022
Apex height (mm)	2.6 (1.5)	2.2 (1.0)	4.2 (1.8)	<0.001
Retinal surface (%)	3.1 (2.4)	2.5 (1.6)	5.0 (3.6)	0.044
T Staging ^‡^:				0.579
T1	9 (25.7%)	7 (27%)	2 (22%)	
T2	26 (74.3%)	19 (73%)	7 (78%)	

* Results expressed as mean (standard deviation) or in absolute frequencies (relative) depending on the nature of the variable. The percentages refer to the total number of cases per column and have been rounded so as not to present decimal values. ^¶^ Calculated by chi-square or Mann-Whitney test according to nature of variable. ^‡^ Staging according to the American Joint Committee on Cancer classification [9].

**Table 2 jcm-11-01616-t002:** Descriptive brachytherapy data according to the initial approach.

Variable	Total Sample * (*n* = 35)	Brachytherapy + Surgery * (*n* = 26)	Brachytherapy Alone * (*n* = 9)	*p* Value ^¶^
Isotope used:				0.013
Ru-106	32 (91.4%)	26 (100%)	6 (66.7%)	
I-125	3 (8.6%)	0 (0%)	3 (33.3%)	
Dose in sclera (Gy)	350.4 (182.9)	294.2 (118.9)	512.6 (241.3)	<0.001
Dose in sclera according to COMS study (Gy) ^†^	247.9 (124.5)	205.5 (73.8)	370.7 (161.5)	<0.001
Optic nerve dose, *n* = 4 (Gy)	12.1 (8.4)	9.3 (13.1)	14.8 (2.9)	0.99
Dose in the macula, *n* = 3 (Gy)	19.9 (4.0)	23.2	18.3 (4.1)	0.67
Lens dose (Gy)	20.3 (14.0)	15.6 (7.9)	33.7 (19.2)	0.023
Dose to opposite retina, *n* = 3 (Gy)	12.4 (1.8)	13.2	12.1 (2.4)	0.99
Outcome after brachytherapy:				
Enucleation	3 (8.6%)	2 (7.7%)	1 (11%)	0.6
Local tumor recurrence	3 (8.6%)	0 (0%)	3 (33.3%)	0.013

* Results expressed as mean (standard deviation) or in absolute frequencies (relative) depending on the nature of the variable. The percentages refer to the total number of cases per column and have been rounded so as not to present decimal values. ^¶^ Calculated by chi-square or Mann-Whitney test according to nature of variable. ^†^ The sclera dose is the dose which comes in contact with the plaque, that is, on the outside of the sclera. The dose assigned by the COMS study is a dose in the sclera applied in the inner part of the sclera, assuming an average thickness of 1 mm.

**Table 3 jcm-11-01616-t003:** Clinical outcomes and complications during follow-up according to the initial approach.

Variable	Total Sample * (*n* = 35)	Brachytherapy + Surgery * (*n* = 26)	Brachytherapy Alone * (*n* = 9)	*p* Value ^¶^
Follow-up time (years)	5.9 (3.0)	5.7 (1.9)	6.5 (5.1)	0.59
Relapse	3 (8.5%)	0 (0%)	3 (33.3%)	0.013
Metastasis	1 (3%)	1 (4%)	0 (0%)	0.74
Initial visual acuity	0.6 (0.2)	0.6 (0.3)	0.6 (0.1)	0.53
Visual acuity at 3 months	0.4 (0.2)	0.4 (0.2)	0.3 (0.3)	0.65
Visual acuity at 6 months	0.3 (0.3)	0.4 (0.3)	0.1 (0.2)	0.06
Visual acuity at 12 months	0.3 (0.3)	0.4 (0.3)	0.1 (0.2)	0.025
Visual acuity at 36 months	0.3 (0.3)	0.3 (0.4)	0.3	0.9
Visual acuity at the end of follow-up	0.3 (0.3)	0.4 (0.3)	0.07 (0.1)	0.07
Complications:				
Scleromalacia	14 (40%)	12 (46%)	2 (22%)	0.26
Bullous keratopathy	11 (31.4%)	9 (34.6%)	2 (22%)	0.68
Retinal toxicity	11 (31.4%)	8 (31%)	3 (33.3%)	0.13
Cataracts	9 (25.7%)	5 (19%)	4 (44.4%)	0.19
Hypotonia	7 (20%)	5 (19%)	2 (22%)	0.59
Photophobia	5 (14.3%)	2 (7.7%)	1 (11%)	0.6
Glaucoma	3 (8.6%)	5 (19%)	0 (0%)	0.29
Infection	1 (3%)	1 (4%)	0 (0%)	0.74
Irreducible retinal detachment	1 (3%)	1 (4%)	0 (0%)	0.74
Others	19 (54.3%)	16 (61.5%)	3 (33.3%)	0.24
Survival:				0.99
At five years	35 (100%)	26 (100%)	9 (100%)	
At seven years	35 (100%)	26 (100%)	9 (100%)	
At 10 years	34 (97%)	25 (96%)	9 (100%)	

* Results expressed as mean (standard deviation) or in absolute frequencies (relative) depending on the nature of the variable. The percentages refer to the total number of cases per column and have been rounded so as not to present decimal values. ^¶^ Calculated by chi-square or Mann-Whitney test according to the nature of the variable.

## Data Availability

The authors confirm that the data supporting the findings of this study are available within the article.

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
