# Peer review of "Clinical Outcomes after Surgical Resection Combined with Brachytherapy for Uveal Melanomas"

_jcm, 2022, doi:10.3390/jcm11061616_

Round 1

Reviewer 1 Report

The authors report on their experience with the combination of brachytherapy and surgical resection of anteriorly-located uveal melanoma. However, in the paper, they subsequently compare 4 cases with brachytherapy with 31 cases with brachytherapy plus surgery. This is not an appropriate comparison. An essential parameter is missing: how many tumors were iris only, how many were iris plus ciliary body and how many ciliary body without iris? In these groups it looks as if there or no more than 6 that do not involve the iris, so that makes comparisons hard. How many were irradiated some time prior to resection? The large ones? They can write up a description for two groups: irradiated at the time of surgery, versus late surgery. The current comparison is not of interest to readers, as this paper was focused on the combination of surgery and irradiation.

A pure description of cases and outcome would also be sufficient.

I advise the authors to look up the literature of the last 5 years, not of over 20 years ago.

Abstract: line 26. Change to: We show that in carefully selected patients, this approach can be effective and safe, ..

Please remove the sentence in the abstract and the introduction about the limited number of papers about surgical uveal melanoma resection. Many papers have been published about removal of iris tumors, and in my opinion, quite a lot about endoresections as well.

The COMS classification is really no longer used. Please remove that part in Methods. For T staging please use the proper 8th edition of the TNM/AJCC classification, not the old one.  The TCGA was developed to compare papers.

Page 3, line 85 esclerectomy? Do you mean sclerectomy?

Line 100. Since in these cases, which were associated with .. : is not proper English. Please rephrase.

Page 4, line 122 hypotony

Page 5, line 124: melanoma cells

Figure 5; the legends and the pictures are not in agreement with each other. . For a: how do you know this is migration of macrophages?

Page 5 line 138: what do dura mater and equine pericardium and tenon have to do with each other?

Line 140: what is septoplasty?

Page 6, line 160: what is sine die?

Line 165: what is graduation? Why did you not correct for astigmatism for determining visual acuity?

Line 181: correct the sequence of sentence

195: please round of percentages to whole numbers (66 instead of 65.7)

Table 1: What do the P values analyse in this descriptive table? You only have 4 cases in the brachytherapy alone group, so this can purely be descriptive, not analytical. Please remove the p values.

please separate anatomical extension into: iris alone

Iris plus ciliary body

Ciliary body without choroid

Ciliary body with choroid.

Table 1: round off percentages to whole numbers, use only 2 digits after the . in non-significant p values.

Indicate which ones are SDs

Remove COMS

Use the proper TNM AJCC classification.

The discussion should be shortened y half and be more focused on discussion points.

Refs 3, 7, 8 are too old, and Damato has written many more recent papers about endoresction than the ones mentioned. Ref 21 has no place here.

Author Response

COMMENT: The authors report on their experience with the combination of brachytherapy and surgical resection of anteriorly-located uveal melanoma. However, in the paper, they subsequently compare 4 cases with brachytherapy with 31 cases with brachytherapy plus surgery. This is not an appropriate comparison. 

ANSWER: We would like to thank the reviewer for the thorough evaluation. We agree that the two groups are unbalanced. Following this and subsequent comments below we have modified the tables and the perspective of the results.

An essential parameter is missing: how many tumors were iris only, how many were iris plus ciliary body and how many ciliary body without iris? In these groups it looks as if there or no more than 6 that do not involve the iris, so that makes comparisons hard. 

ANSWER: We thank the reviewer for this comment. We have now completed the table with the local extension as suggested by the reviewer

How many were irradiated some time prior to resection? The large ones? They can write up a description for two groups: irradiated at the time of surgery, versus late surgery. The current comparison is not of interest to readers, as this paper was focused on the combination of surgery and irradiation. A pure description of cases and outcome would also be sufficient.

ANSWER: We appreciate this comment and would like to thank for allowing us to show the information in a more proper way. We have now classified the results as suggested by the reviewer and modified the tables and the results accordingly.

COMMENT: I advise the authors to look up the literature of the last 5 years, not of over 20 years ago.

ANSWER: We have reviewed the references and updated them, as suggested.

COMMENT: Abstract: line 26. Change to: We show that in carefully selected patients, this approach can be effective and safe, ..

ANSWER: We have modified the sentences, as per the suggestion.

COMMENT: Please remove the sentence in the abstract and the introduction about the limited number of papers about surgical uveal melanoma resection. Many papers have been published about removal of iris tumors, and in my opinion, quite a lot about endoresections as well.

ANSWER: We have eliminated the sentences, as per the suggestion.

COMMENT: The COMS classification is really no longer used. Please remove that part in Methods. For T staging please use the proper 8th edition of the TNM/AJCC classification, not the old one.  The TCGA was developed to compare papers.

ANSWER: We have removed the references to old classifications and we have reclassified patients according to the more recent TNM /AJCC classification

COMMENT: Page 3, line 85 esclerectomy? Do you mean sclerectomy?

ANSWER: Thanks for noticing this typo, now corrected

COMMENT: Line 100. Since in these cases, which were associated with .. : is not proper English. Please rephrase.

ANSWER: We agree with the reviewer. We have modified the wording.

COMMENT: Page 4, line 122 hypotony

ANSWER: Thanks for noticing this typo, now corrected. We have searched PubMed and found both hypotonia and hypotony equally used. We have changed it to hypotonia, but we will be happy to use the term that the reviewer considers more correct.

COMMENT: Page 5, line 124: melanoma cells

ANSWER: Thanks for noticing this typo, now corrected

COMMENT: Figure 5; the legends and the pictures are not in agreement with each other. . For a: how do you know this is migration of macrophages?

ANSWER: We apologize for the confusion. Figures 5a and 5f were wrongly placed. Now corrected.

COMMENT: Page 5 line 138: what do dura mater and equine pericardium and tenon have to do with each other?

ANSWER: We agree with the reviewer. We have modified the wording.

COMMENT: Line 140: what is septoplasty?

ANSWER: Thanks for noticing this typo. We refer to conjunctival zetaplasty.

COMMENT: Page 6, line 160: what is sine die?

ANSWER: Sine die is a latin expression meaning without any future date being designated or indefinitely. We have changed the expression.

COMMENT: Line 165: what is graduation? Why did you not correct for astigmatism for determining visual acuity?

ANSWER: We have modified the wording to make the text clearer.

COMMENT: Line 181: correct the sequence of sentence

ANSWER: We have modified the sequence, as per the suggestion.

COMMENT: 195: please round of percentages to whole numbers (66 instead of 65.7)

ANSWER: We are not sure we understand this comment from the reviewer. We don't know if he/she wants us to change the decimals of this particular percentage or if what he/she is suggesting is that we remove all the decimals from the manuscript. We have used a single decimal consistently throughout the document in order to give more exact figures. We have now eliminated the number of decimal points, as per the suggestion. 

COMMENT: Table 1: What do the P values analyse in this descriptive table? You only have 4 cases in the brachytherapy alone group, so this can purely be descriptive, not analytical. Please remove the p values.

ANSWER: We have modified all tables in the manuscript according to this and previous comments from the reviewer. Our intention is to inform the scientific community of the results and complications of these procedures, with a special focus on cases in which surgery and brachytherapy are performed in the same act. Therefore, we have divided the data comparing this procedure with other approaches such as brachytherapy alone or followed by surgery.

COMMENT: please separate anatomical extension into: iris alone, Iris plus ciliary body, Ciliary body without choroid, Ciliary body with choroid.

ANSWER: We have modified this information in table 1, as per the suggestion.

COMMENT: Table 1: round off percentages to whole numbers, use only 2 digits after the . in non-significant p values. Indicate which ones are SDs. Remove COMS. Use the proper TNM AJCC classification.

ANSWER: We have removed the COMS classification and used the AJCC one. As for the decimal, we are not sure we understand this comment from the reviewer. We understand that the reviewer wants us to remove all decimals from all percentage values and leaving the decimals in the quantitative variables, except for the p-values, which we should leave with 3 decimal places if they are significant and with 2 decimal places if they are not. Is this correct? It is generally well appreciated to be consistent in the way the numerical results of a paper are reported, and editors often prefer that the same number of decimal places be used throughout the document. Additionally, rounding to whole numbers without decimals decreases the accuracy of the data. As indicated by the reviewer, we have made these changes.

COMMENT: The discussion should be shortened by half and be more focused on discussion points.

ANSWER: We have reviewed the discussion trying to focus it better. However, we would appreciate if the reviewer could clarify what aspects of the discussion are not relevant and therefore would like us to eliminate.

COMMENT: Refs 3, 7, 8 are too old, and Damato has written many more recent papers about endoresction than the ones mentioned. Ref 21 has no place here.

ANSWER: We have corrected reference 21 citation and updated references.

Reviewer 2 Report

The paper is nicely written but suffers because of the design, given the single center data and being retrospective. 

The introduction is clear and the description of the surgical technique excellent.

The data, although presented in 3 tables, according my opinion should be more clarified, ie A table with the population under exam together with the parameters under investigation should be presented first. the univariate and the multivariate analysis should follow and the results discussed. 

I do not really understand how 16 men in 35 patients mean that there is a predominance which sounds a bit strange since 3 female patients more, do not signify a female predominance in a 35 patient sample.

I think that if the authors  put an extra data table with just numeric and/or percentage values and then present their multivariate eller univariate analyses, it would be easier for the reader.

Plus I would like to see some more about systemic treatments and although there is nothing applied yet, have authors opinion on immunotherapy eller other agents in the treatment of the primary tumor.

Overall it is an interesting paper and although single center and retrospective, can be published if these minor changes could be made.

Author Response

COMMENT: The paper is nicely written but suffers because of the design, given the single center data and being retrospective. 

ANSWER: We agree with the reviewer, and we have included these limitations in the discussion.

COMMENT: The introduction is clear and the description of the surgical technique excellent.

ANSWER: We would like to thank the reviewer for the positive comment and the thorough evaluation.

COMMENT: The data, although presented in 3 tables, according my opinion should be more clarified, ie A table with the population under exam together with the parameters under investigation should be presented first. the univariate and the multivariate analysis should follow and the results discussed. 

ANSWER: Following this comment and the comments from reviewer #1 we have modified the tables and the results to give a clearer picture of our findings.

COMMENT: I do not really understand how 16 men in 35 patients mean that there is a predominance which sounds a bit strange since 3 female patients more, do not signify a female predominance in a 35 patient sample.

ANSWER: The reviewer is correct. We have modified the description of the cases in the results section, as per the suggestion.

COMMENT: I think that if the authors  put an extra data table with just numeric and/or percentage values and then present their multivariate eller univariate analyses, it would be easier for the reader.

ANSWER: Following this comment and the comments from reviewer #1 we have modified the tables and the results to give a clearer picture of our findings.

COMMENT: Plus I would like to see some more about systemic treatments and although there is nothing applied yet, have authors opinion on immunotherapy eller other agents in the treatment of the primary tumor.

ANSWER: This is an interesting and pertinent comment and question. Unfortunately, we did not record systemic therapies in the dataset. 

COMMENT: Overall it is an interesting paper and although single center and retrospective, can be published if these minor changes could be made.

ANSWER: We would like to thank the reviewer for the positive comment. We have answered all queries and we hope you now have sufficient information to complete a positive editorial decision.

Round 2

Reviewer 1 Report

The authors did a fine job, using the comments to improve the paper nicely.

Just two technical comments:

1. it reads better when % is added in the tables where appropriate.(Table 1, 2, 3)

2. what I meant is that there should consistentely be 2 digits with percentages, unless it is 100 when there are three. A percentage of 11.1% does not say more than a percentage of 11%:   17.1 should become 17 % while 7.7 can remain the same, while written as 7.7 %. it should be concistent in all tables.

Author Response

The authors did a fine job, using the comments to improve the paper nicely. 

Just two technical comments:

COMMENT 1. it reads better when % is added in the tables where appropriate.(Table 1, 2, 3)

ANSWER: Thanks for the evaluation. We have added the % symbol in all tables.

COMMENT 2. what I meant is that there should consistentely be 2 digits with percentages, unless it is 100 when there are three. A percentage of 11.1% does not say more than a percentage of 11%:   17.1 should become 17 % while 7.7 can remain the same, while written as 7.7 %. it should be concistent in all tables.

ANSWER: We have reviewed all percentage values in all tables. Following these instructions, when the decimal value is 0.7, 0.6, 0.5, 0.4, 0.3 we have left the decimal as it is. On the other hand, if the decimal value is 0.9, 0.8, 0.2, 0.1 we have rounded the value to the whole number. I hope we have understood the instructions correctly.